

# Short-term power load forecasting based on gray relational analysis and support vector machine optimized by artificial bee colony algorithm

Xinfu Pang[1], Wei Sun[1], Haibo Li[1], Yibao Wang[1] and Changfeng Luan[2]

[1] Key Laboratory of Energy Saving and Controlling in Power System of Liaoning Province, Shenyang Institute of Engineering, Shenyang, Liaoning, China
[2] Yingkou Power Supply Company, State Grid Liaoning Electric Power Co., Ltd., Yingkou, Liaoning, China

## ABSTRACT

Short-term power load forecasting is essential in ensuring the safe operation of power systems and a prerequisite in building automated power systems. Short-term power load demonstrates substantial volatility because of the effect of various factors, such as temperature and weather conditions. However, the traditional short-term power load forecasting method ignores the influence of various factors on the load and presents problems of limited nonlinear mapping ability and weak generalization ability to unknown data. Therefore, a short-term power load forecasting method based on GRA and ABC-SVM is proposed in this study. First, the Pearson correlation coefficient method is used to select critical influencing factors. Second, the gray relational analysis (GRA) method is utilized to screen similar days in the history, construct a rough set of similar days, perform $K$-means clustering on the rough sets of similar days, and further construct the set of similar days. The artificial bee colony (ABC) algorithm is then utilized to optimize penalty coefficient and kernel function parameters of the support vector machine (SVM). Finally, the above method is applied on the basis of actual load data in Nanjing for simulation verification, and the results show the effectiveness of the proposed method.

# INTRODUCTION

Short-term power load forecasting is the basis for the safe operation of power systems and the formulation of dispatching plans (*Liao et al., 2011*). The cyclical nature and substantial uncertainty in the load due to the effect of many factors lead to many load forecasting challenges. Short-term power load forecasting based on selecting similar days can provide a reference for the operation and dispatch of regional power systems.

Short-term load forecasting approaches are mainly categorized as traditional and machine learning methods. Traditional methods include the time series (*Borges, Penya & Fenandez, 2013*; *Espinoza et al., 2005*), Kalman filter (*Sharma & Majumdar, 2020*), exponential smoothing (*Arora & James, 2013*) methods and so on. Machine learning

Corresponding author
Xinfu Pang, pangxf@sie.edu.cn

methods include artificial neural networks (*Deng et al., 2019*; *Liu, Liu & Zhang, 2022*), support vector machine (*Chen, Chang & Lin, 2004*; *Qi et al., 2021*), long short-term memory networks (*Rafi et al., 2021*) and so on. *Mai et al. (2015)* presents a short-term load forecasting method based on the Kalman filter, but its weak mapping ability to nonlinearity affects the forecasting accuracy. *Hu, Guo & Wang (2021)* uses sparrow search algorithm to optimize the penalty coefficient $C$ and kernel function parameter $g$ of the least squares support vector machine and improve the prediction accuracy. *Wang, Dou & Meng (2021)* utilizes a multicore extreme learning machine with improved particle swarm optimization parameters for prediction. However, the global search ability is weak when particle swarm optimization parameters are optimized. *Liu et al. (2018)* applies empirical mode decomposition to divide the load into various IMS components and bat algorithm to optimize parameters of the support vector machine. The output results are then corrected using the Kalman filter method to improve the prediction accuracy. However, the bat algorithm demonstrates the limitations of long search time and weak global search ability. To sum up, although the traditional method based on statistics presents advantages of simplicity and speed, it ignores the influence of various external factors on the load and shows poor nonlinear mapping ability. The method based on machine learning is suitable for processing. The nonlinear short-term power load problem exhibits strong generalization ability to anonymous data but difficulty in optimizing parameters.

Preprocessing historical data is essential before using them as input data for short-term load forecasting. *Zhu et al. (2021)*, *Ceperic, Ceperic & Baric (2013)*, *Chen et al. (2021)*, *Farsi et al. (2021)*, *Pham, Nguyen & Wu (2021)* and *Yin et al. (2020)* screen external factors that affect the load but fail to select similar days for historical data; the results showed that massive data unrelated to the day to be forecasted in the input data occupy a large amount of computing resources. The forecasting time is long, and the forecasting model fails to predict the causal relationship between external factors and the load on the day to be forecasted accurately. *Zhao & Dai (2020)* uses hourly granularity to select similar days in holiday forecasting to enhance the prediction accuracy but only selects similar days once, only considers the time factor, and ignores important factors, such as weather and temperature. *Liu & Wei (2020)* utilizes an improved gray relational analysis method to select the set of similar days but only considers the geometric similarity between influencing factors and neglects the numerical similarity. Hence, a large error exists in the selection of similar days. *Wu et al. (2018)* adopts an improved gray relational method to select similar days while solving the problem that gray relational analysis only considers the geometric similarity between factors. However, the selection of similar days is not carried out twice, and the collection of similar days is slightly rough. Therefore, the problem of redundant input data after selecting influencing factors and processing a similar day on input data still exists, thereby decreasing the prediction accuracy.

To sum up, the traditional forecasting method ignores the influence of various factors on the short-term power load. Meanwhile, the support vector machine can learn the relationship between external factors and the load and rapidly deal with nonlinear problems. The grey relational analysis method screens historical data for geometric similarity, screening daily historical information similar in shape to the load curve of the

day to be forecasted. In addition, because this method cannot screen daily historical data identical to the daily load value to be predicted, the set of similar days screened out is relatively rough. However, $K$-means clustering filters historical information for numerical similarity. Therefore, using the grey relational analysis method and $K$-means clustering to select two similar days can screen historical data from both geometric and numerical aspects and remove and predict to the greatest extent. A more accurate collection of similar days can be obtained for historical data with low daily correlation. The artificial bee colony algorithm presents the advantages of fast optimization of parameters and difficulty in falling into the local optimum. On this basis, GRA and ABC-SVM short-term power load forecasting methods are proposed. First, the Pearson correlation coefficient method is adopted to screen external factors affecting the load. Second, historical data are screened twice using gray relational analysis and $K$-means clustering. Third, the ABC-SVM load forecasting model is constructed. Finally, actual electricity consumption data of a city for 1 year is verified, and the prediction results are then compared with those of multiple models, such as long short-term memory (LSTM) neural network, to confirm the effectiveness of the proposed method. The proposed method uses grey relational analysis and $K$-means clustering to select similar days twice, which solves the problem of reducing prediction accuracy caused by excessive input of irrelevant data in short-term load forecasting. The artificial bee colony algorithm is used to optimize the critical parameters of the support vector machine model, which avoids the problem that the prediction accuracy is reduced due to improper parameter selection of the support vector machine in short-term load forecasting.

## PROBLEM DESCRIPTION OF SHORT-TERM POWER LOAD FORESCASTING

Short-term power load forecasting must consider not only the change of load over time and the influence of temperature, weather conditions, wind direction, wind force, and other factors on the load but also external factors to improve the prediction accuracy. Hence, identifying external factors that exert a important influence on the load is necessary. At the same time, historical data irrelevant to or weakly correlated with the date to be forecasted are necessary due to the massive amount of data required for short-term load forecasting. If these data are used as input, then the forecasting accuracy will decrease. Therefore, filtering historical data and finding historical data closely related to the day to be predicted are necessary for training. Traditional methods based on statistics are unsuitable for nonlinear problems. Hence, choosing an appropriate machine learning algorithm for short-term power load forecasting is necessary. The problem description of short-term power load forecasting is shown in Fig. 1. your materials and methods here.

## STRATEGY STRUCTURE OF SHORT-TERM POWER LOAD FORECASTING

The process of short-term power load forecasting based on GRA and ABC-SVM is presented as follows. First, the Pearson correlation coefficient analysis method is used to identify

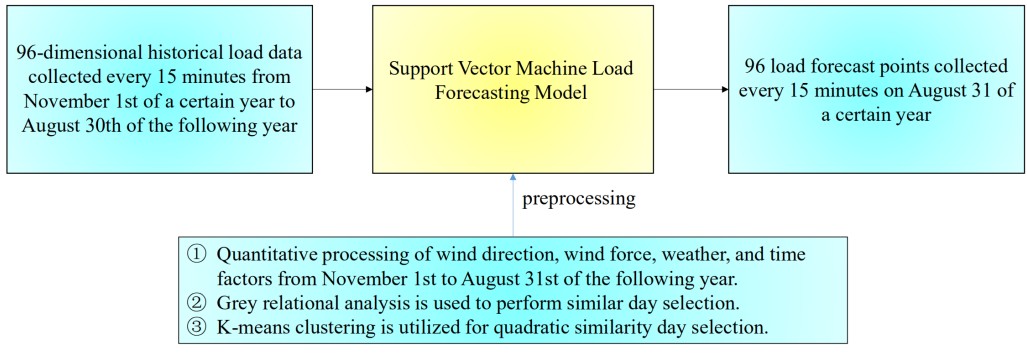

**Figure 1  Description of the short-term power load forecasting problem.**

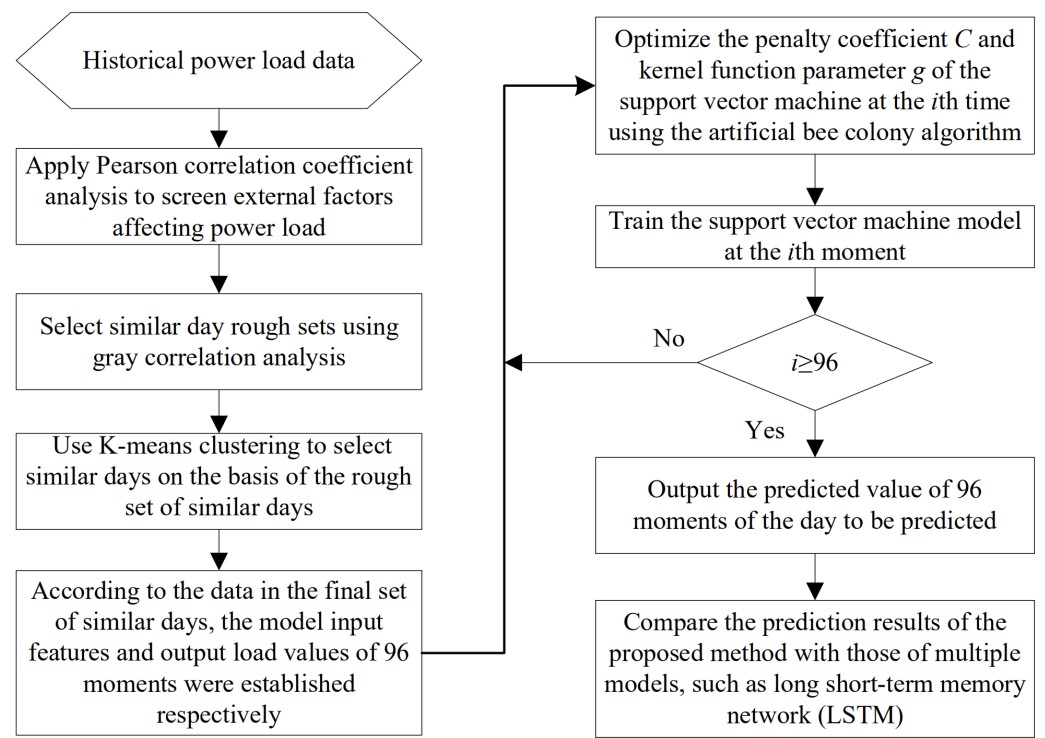

**Figure 2  Strategy map for short-term power load forecasting methods.**

critical influencing factors. Second, the gray correlation analysis method is applied to construct a rough set of similar days, and $K$-means clustering is utilized to filter the similarities again. Third, the ABC algorithm is adopted to find the optimal penalty coefficient $C$ and kernel function parameter $g$ of the SVM. Finally, the similar day set is inputted into the optimized SVM model as input data for training. Mean absolute (MAE), mean absolute percentage (MAPE), and root mean square (RMSE) errors were selected to measure the performance of the forecasting method. The specific implementation strategy of the method is presented in Fig. 2.

**Table 1  Correlation coefficient and correlation degree table.**

| Correlation coefficient interval | Relevance |
|---|---|
| $0.2 \leq |\rho| < 0.4$ | Very weak correlation |
| $0.2 \leq |\rho| < 0.4$ | Weak correlation |
| $0.4 \leq |\rho| < 0.6$ | Moderately relevant |
| $0.6 \leq |\rho| < 0.8$ | Strong correlation |
| $0.8 \leq |\rho| < 1.0$ | Very strong correlation |

## Selection of influencing factors based on Pearson correlation coefficient method

Short-term power load forecasting requires not only massive load data but also external factor data. External factors include temperature, weather conditions, wind direction, wind force, whether the day is a workday, and whether the day is a holiday. Changes in external factors will cause variations in the power load. Therefore, the relationship between load and external factors can be accurately determined by filtering external factors with a significant impact on the load to improve the prediction accuracy.

The Pearson correlation coefficient method is a statistical indicator used to measure the degree of correlation between the influencing factor $X$ and the load $Y$, and its value is in the $[-1,1]$ interval (*Kong & Nian, 2021*). The Pearson correlation coefficient is calculated as follows:

$$\rho = \frac{\sum_{k=1}^{n} (y_k - \bar{y})(x_k - \bar{x})}{\sqrt{\sum_{k=1}^{n} (y_k - \bar{y})^2 \sum_{k=1}^{n} (x_k - \bar{x})^2}}. \tag{1}$$

The degree of correlation between the two variables is high when the absolute value of $\rho$ is close to 1. By contrast, the degree of correlation between the two variables is low when the absolute value of $\rho$ is close to 0.

The standard definition of correlation for the Pearson correlation coefficient analysis method is presented in Table 1.

## Construction of similar daily rough sets based on gray relational analysis

Gray relational analysis is a statistical method for mining the similarity between values (*Kong, Li & Zheng, 2020*). A large correlation value corresponds to high similarity between the historical day and the day to be predicted in the selection process of similar days. By contrast, a small correlation value indicates low similarity between the historical day and the day to be measured. The input is the characteristic value of the maximum temperature, minimum temperature, whether the day is a workday, and whether the day is a holiday in the historical day and the day to be predicted. The output is the gray correlation value between the historical day and the day to be predicted. The process of GRA is illustrated in Fig. 3.

(1) Extract the value of each external factor on the $i$ th day to form a set $Y_i$:

$$Y_i = [y_{i1} y_{i2} y_{i3} \ldots y_{im}], \tag{2}$$

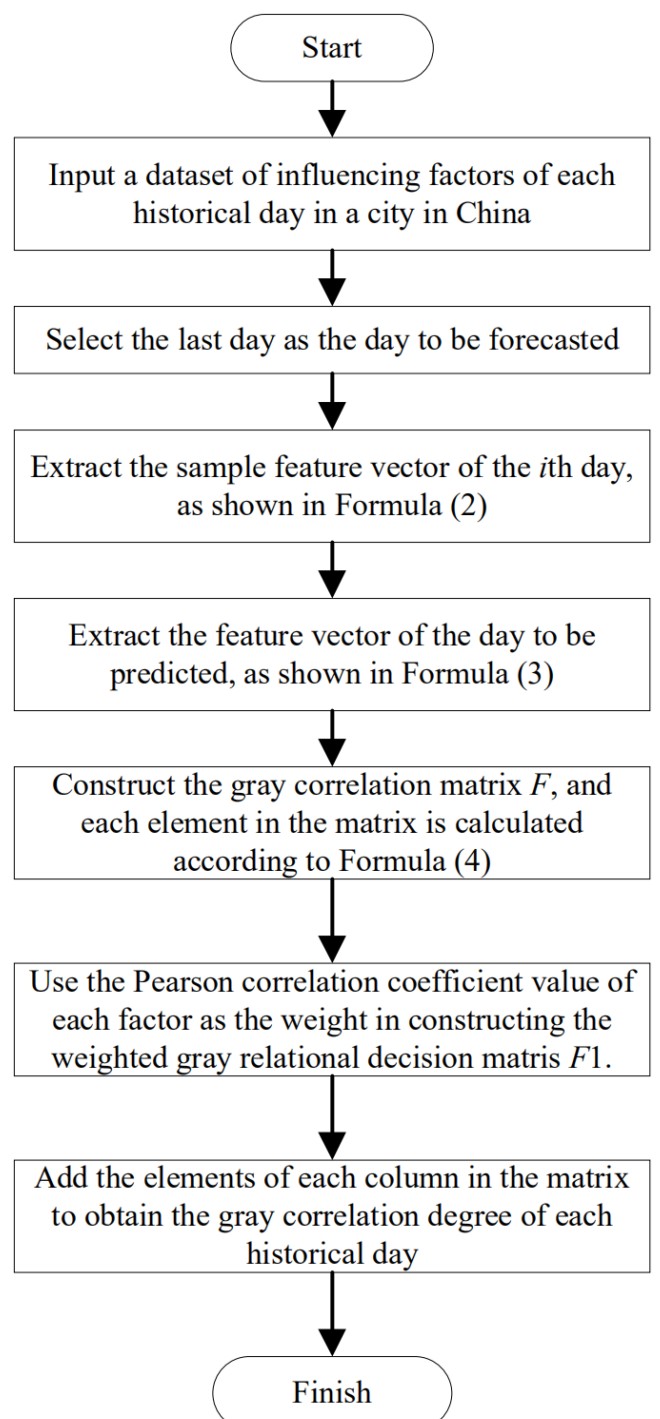

**Figure 3** Flowchart of selecting the similar day rough set through gray relational analysis.

where $i = 1, 2, \ldots, n$; $n$ is the number of historical load data samples; $y_{im}$ is the value of the $m$ th external factor on the $i$ th day; and $m$ is the number of external factors.

Eigenvector of the day to be predicted $Y_0$ is expressed as follows:

$$Y_0 = [y_{01} y_{02} y_{03} \ldots y_{0m}], \tag{3}$$

where $y_{0m}$ is the value of the $m$ th external factor on the day to be predicted.

(2) Construct the gray relational matrix $F$:

$$F = \begin{bmatrix} F_{11} & F_{12} & \ldots & F_{1m} \\ F_{21} & F_{22} & \ldots & F_{2m} \\ . & . & & . \\ . & . & & . \\ F_{n1} & F_{n2} & \ldots & F_{nm} \end{bmatrix}. \tag{4}$$

Each element in the matrix is calculated as follows:

$$F_{ik} = \frac{\min_{i=1,\ldots,n} \min_{k=1,\ldots,m} |y_{0k} - y_{ik}| + \rho \max_{i=1,\ldots,n} \max_{k=1,\ldots,m} |y_{0k} - y_{ik}|}{|y_{0k} - y_{ik}| + \rho \max_{i=1,\ldots,n} \max_{k=1,\ldots,m} |y_{0k} - y_{ik}|}, \tag{5}$$

where $F_{ik} (i = 1, 2, \ldots, n; k = 1, 2, \ldots, m)$ is the correlation coefficient corresponding to the $k$ th external factor of the $i$ th historical load sample, $y_{ik} (i = 1, 2, \ldots, n; k = 1, 2, \ldots, m)$ is the value of the $k$ th external factor of the $i$ th historical load sample, $y_{0k} (k = 1, 2, \ldots, m)$ is the value of the $i$ th external factor on the day to be forecasted, and $\rho$ is the resolution coefficient typically set to $\rho = 0.5$. The Pearson correlation coefficient method is applied to determine the weight of each external factor as follows:

$$W = [\omega_1, \omega_2, \ldots, \omega_m], \omega_k = \frac{p_k}{\sum_{k=1}^{m} p_k}, \tag{6}$$

where $p_k$ is the absolute value of the Pearson correlation coefficient of the $k$ th external factor; $\omega_k$ is the weight occupied by the $k$ th external factor; and $k = 1, 2, \ldots, m$.

(3) Use the weights of external factors to weigh the gray relational matrix $F$ and construct the gray relational decision matrix $F_1$:

$$F_1 = FW^T = \begin{bmatrix} \omega_1 F_{11} & \omega_2 F_{12} & \ldots & \omega_m F_{1m} \\ \omega_1 F_{21} & \omega_2 F_{22} & \ldots & \omega_m F_{2m} \\ . & . & & . \\ . & . & & . \\ \omega_1 F_{n1} & \omega_2 F_{n2} & \ldots & \omega_m F_{nm} \end{bmatrix}. \tag{7}$$

(4) Add the element values of each row in the matrix to obtain the correlation value $D_i$ of each historical sample.

$$D_i = \sum_{k=1}^{m} \omega_k F_{ik}, \tag{8}$$

where $D_i$ is the gray correlation value between the $i$ th historical day and the day to be predicted.

## Secondary selection of similar day sets based on *K*-means clustering

*K*-means clustering (*Xi et al., 2019*) is used to build a set of similar days to reduce historical data in the set with low correlation and days to be predicted further given that the gray relational analysis method only builds a rough set of similar days for geometric similarity. First, the number of cluster centers is set by the silhouette coefficient. Second, the Euclidean distance between influencing factors and the cluster center in each historical day sample in the rough set of similar days is calculated using Formula (9), and the daily samples are classified. Finally, the distance between each cluster center and influencing factors of the day to be predicted is calculated, the cluster center with the minimum distance is selected, and the historical daily samples included in the cluster center are considered the final set of similar days. This process is shown in Fig. 4.

$$d_j = \sqrt{\sum_{k=1}^{m} (x_k - p_{jk})^2}, \tag{9}$$

where $x_k \, (k = 1, 2, \ldots, m)$ is the eigenvalue of the $k$ th factor of influencing factors on the day to be predicted and $p_{jk} \, (j = 1, 2, \ldots, l; k = 1, 2, \ldots, m)$ is the value of the $k$ th influence factor of the $j$ th group of cluster centers.

## ABC-SVM short-term power load forecasting model
### *Mathematical model of ABC algorithm to optimize SVM parameters*

(1) Objective function

The minimum mean square error $J$ between predicted and actual load values of the support vector machine is used as the objective function in this study.

$$\min J = \frac{1}{N} \sum_{i=1}^{N} (y_i' - y_i)^2, \tag{10}$$

where $N$ is the time point, $y_i'$ is the predicted load value at the $i$ th time, and $y_i$ is the actual load value at the $i$ th time.

(2) Constraints

Setting upper and lower bound constraints on parameters to be optimized is necessary to improve the accuracy of the algorithm, find optimal parameters of the support vector machine, and reduce the optimization time of the algorithm.

$$\begin{cases} C \in [0.01, 50] \\ g \in [0.01, 50] \end{cases}, \tag{11}$$

where $C$ is the penalty coefficient of the support vector machine and $g$ is the kernel function parameter of the support vector machine.

### *Design of artificial bee colony optimization algorithm*

*Karaboga & Akay (2007)* proposed an intelligent optimization algorithm that imitates the honey-collecting behavior of bees called the artificial bee colony (ABC) algorithm in 2005. ABC consists of four parts, namely, food source, lead bee, watcher bee, and scout bee, and

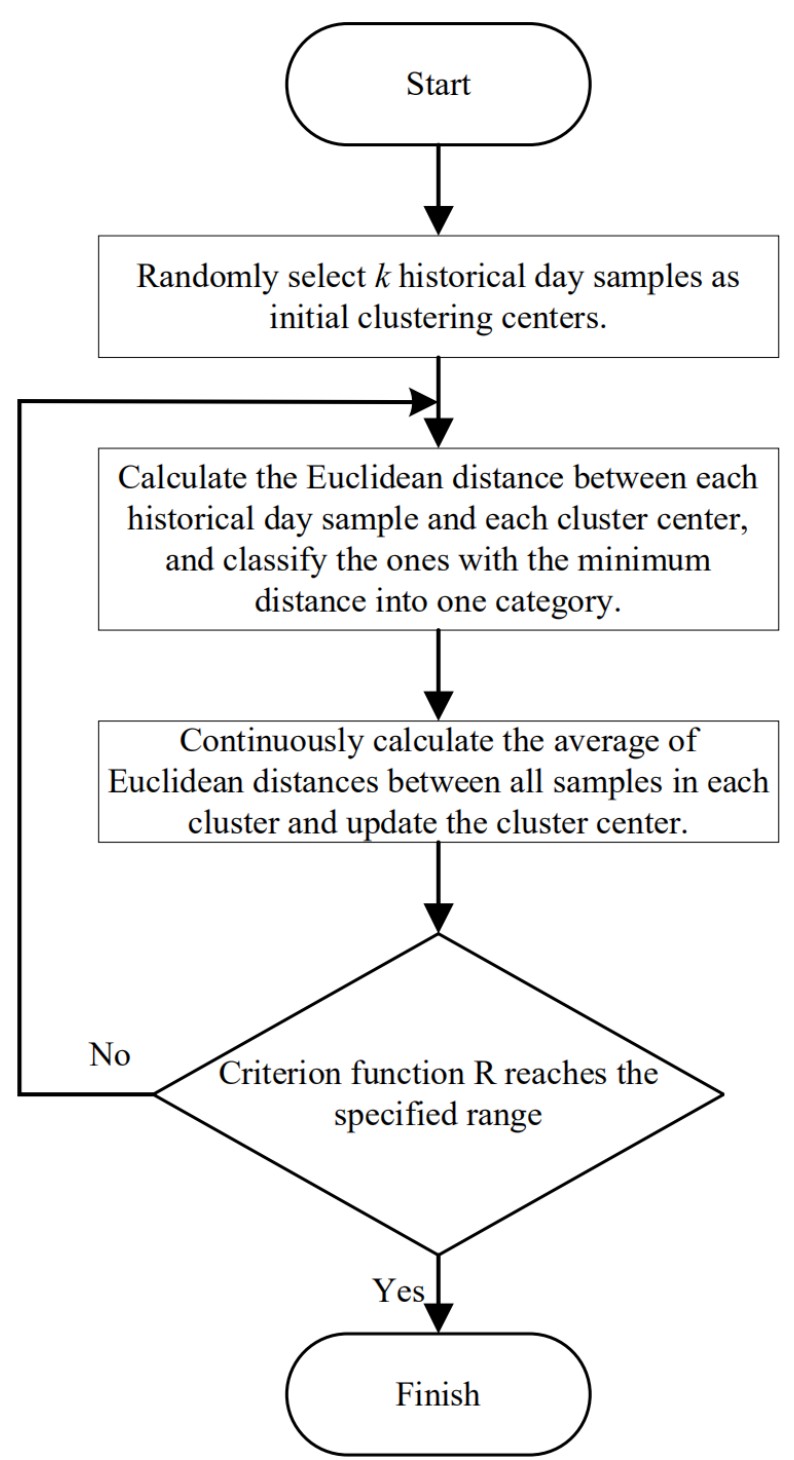

**Figure 4** *K*-means clustering flowchart for selecting similar day sets.

**Table 2  Correspondence between honeybee collecting behavior and parameter optimization problem.**

| Honeybee collection behavior | Specific optimization problems |
| --- | --- |
| Food source | Two-dimensional vector composed of SVM parameters $C$ and $g$ |
| Amount of nectar from food source | Fitness value in the problem |
| Process of finding and gathering food sources | SVM prediction process |
| Food source for the maximum amount of nectar | Optimal parameters $C$ and $g$ of SVM |

demonstrates the advantages of fast optimization and difficulty in falling into the local optimum.

The relationship between the honeybee collection behavior and the optimization problem in the ABC algorithm is presented in Table 2.

The process of finding optimal parameters of the SVM is presented as follows:

(1) Initialize the honey source

First, set the population size, number of feasible solutions $SN$, maximum number of iterations $MCN$, control parameter *limit*, upper and lower bounds of the penalty coefficient $C$, and kernel function parameter $g$. Second, randomly generate two-dimensional $SN$ vector solutions $X_i^t$ $(i = 1, 2, \ldots, SN)$ according to Formula (12), where $t$ is the number of iterations.

$$x_{i,j} = x_{\min,j} + rand\,(0, 1)\,(x_{\max,j} - x_{\min,j}), \tag{12}$$

where $j \in (1, 2)$ and $x_{max,j}$ and $x_{min,j}$ are the upper and lower bounds of the penalty coefficient $C$ and the kernel function parameter $g$, respectively. These feasible solutions will be randomly assigned to $SN$-employed bees, and the fitness of these solutions will be calculated. The relationship between the objective function value of the solution and the fitness value is expressed as follows:

$$fit_i = \begin{cases} 1/(1+f_i), & f_i \geq 0 \\ 1 + \mathrm{abs}\,(f_i), & otherwise \end{cases} \tag{13}$$

where $f_i$ is the objective function value.

The objective function value takes the feasible solution as the penalty coefficient $C$ and the kernel function parameter $g$ of the SVM. The mean square error between predicted and actual load values of the predicted day is obtained after inputting historical data in a similar day set for training.

(2) Leading bee stage

Each lead bee uses Formula (14) to search around its feasible solutions and find new feasible solutions. The mean square error between predicted and actual load values of the support vector machine corresponding to the new solution will be calculated when a new solution is found. If the mean square error corresponding to the new solution is smaller than the old solution, then the lead bee will give up the old solution. A new solution is selected and the penalty coefficient $C$ and kernel function parameter $g$ corresponding to

the new solution are recorded. Conversely, the lead bee retains the old solution.

$$v_{i,j} = x_{i,j} + \varphi_{i,j}(x_{i,j} - x_{k,j}), \tag{14}$$

where $k \in \{1, 2, \ldots, SN\}$ and $j \in (1, 2)$ are randomly selected values in the corresponding interval, $k \neq i$, and $\varphi_{i,j}$ is a random number in the range $[-1, 1]$.

(3) Calculate the probability that the follower bee follows the leader bee

Leading bees will dance in the recruitment area to share the solution information with the follower bees after all the leading bees complete the search. The follower bees calculate the selection probability of each solution as follows:

$$p_i = \frac{fit_i}{\sum_{i=1}^{SN} fit_i}, \tag{15}$$

where $fit_i$ isthe fitness value of the $i$ th solution.

(4) Scouting bee search stage

The scout bee stage prevents the ABC algorithm from falling into the local optimum. If a feasible solution $X_i^t$ has not been updated, then the number of times *the trail* exceeds the *limit*. The hired bee corresponding to the solution will become a scout bee, abandon the solution, and find a new solution in the neighborhood of the solution. The new solution $X_i^{t+1}$ is determined as follows:

$$X_i^{t+1} = \begin{cases} x_{min,j} + rand \cdot (x_{max,j} - x_{min,j}), trail \geq limit \\ X_i^t, trail_i < limit \end{cases} \tag{16}$$

Figure 5 shows the flowchart of the ABC algorithm to express the specific steps of the ABC algorithm clearly for optimizing the parameters of the support vector machine.

### SVM short-term power load forecasting model

SVM is a machine learning algorithm proposed by Vapnik et al. based on the structural risk minimization criterion in statistical learning theory, which has the functions of classification and regression (*Madhukumar et al., 2022*; *Li et al., 2020*; *Tan et al., 2020*). Short-term load forecasting uses the regression function of the support vector machine. The short-term power load presents nonlinear characteristics because of the effect of weather, temperature, and time. Hence, finding a general rule is difficult. SVM can map the short-term power load that shows nonlinear laws from the low-dimensional space to the high-dimensional space through the kernel function and then predicts the load.

Specifically, the training set is a collection of $m$ historical daily sample data and expressed as follows:

$$T = \{(x_1, y_1), (x_2, y_2), \ldots, (x_m, y_m)\} \in (X \times Y)^m, \tag{17}$$

where $x$ is the normalized value of the highest and lowest temperatures in the historical day, which may a workday or a holiday; $y$ is the load value of the historical day; and $m$ is the number of samples in the training set.

SVM regression is ideal for solving the load forecasting problem and obtaining a regression model (Formula (18)). Note that the predicted value $f(x)$ is close to the actual

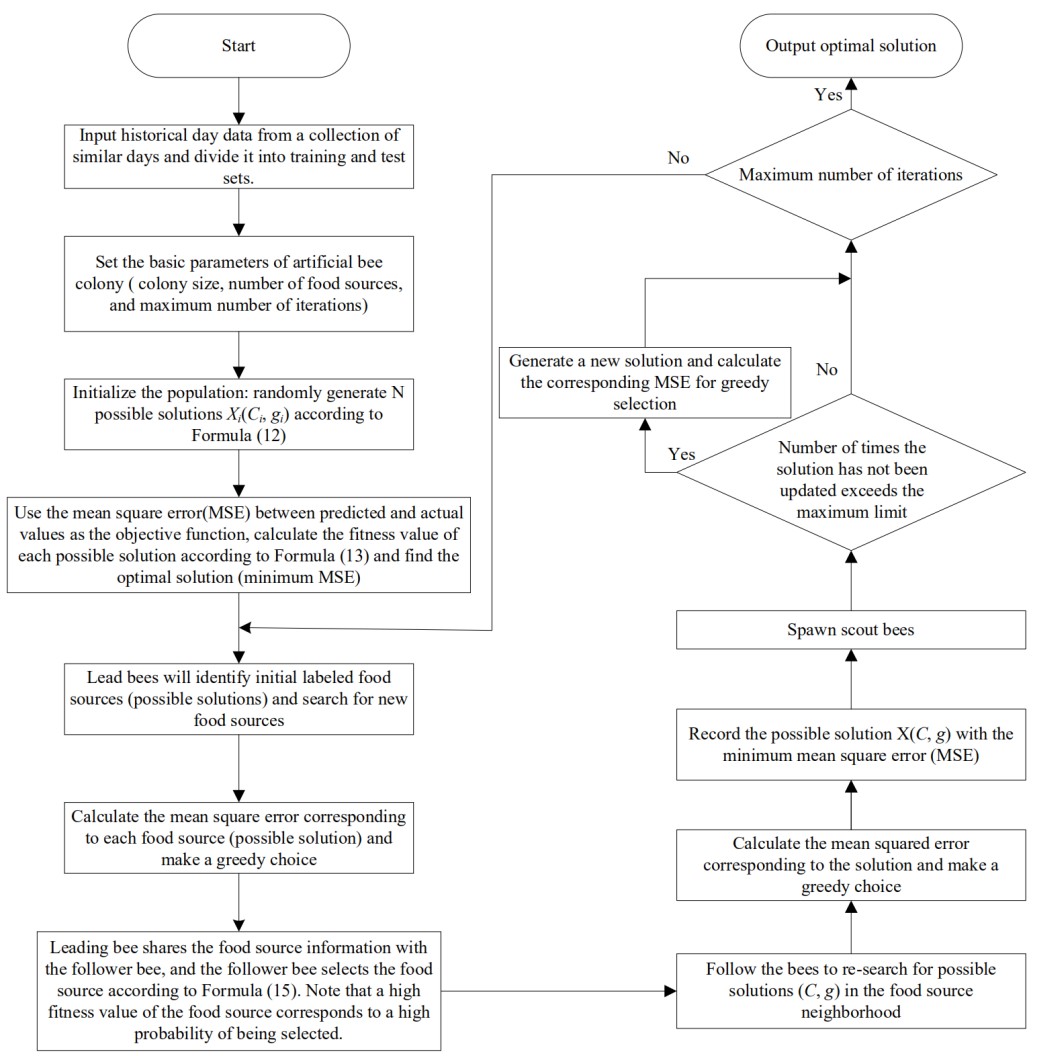

**Figure 5** Flowchart of ABC algorithm in optimizing SVM parameters.

load value $y$.

$$f(x) = \omega^T \varphi(x) + \hat{b}, \tag{18}$$

where $\omega$ is the weight vector, $b$ is a constant, and $\varphi(x)$ is the nonlinear mapping function.

SVM can be expressed after adding the slack variable $\xi_i, \hat{\xi}_i$ and penalty coefficient $C$ to Eq. (18) as follows:

$$\begin{cases} \min \dfrac{1}{2}\|\omega\|^2 + C \displaystyle\sum_{i=1}^{m} \left( \xi_i - \hat{\xi}_i \right) \\ \text{s.t.} f(x_i) - y_i \le \varepsilon + \hat{\xi}_i \\ y_i - f(x_i) \le \varepsilon + \hat{\xi}_i \\ \xi_i \ge 0, \hat{\xi} \ge 0, i = 1, 2, \ldots, m \end{cases} \tag{19}$$

where $\varepsilon$ is an insensitive loss factor. The insensitive loss function determines the acceptable error size during SVM training. The larger $\varepsilon$ is, the greater the acceptable error during SVM training, and the lower the prediction accuracy. The smaller $\varepsilon$, the smaller the acceptable error during SVM training, and the lower the prediction accuracy. If $\varepsilon$ is too large, it will lead to under-fitting in training, and if $\varepsilon$ is too small, it will lead to over-fitting in training, so the default value of $\varepsilon = 0.1$ is usually taken. The dual problem of SVM can be obtained by introducing Lagrange multipliers.

$$\max_{\alpha,\hat{\alpha}} \sum_{i=1}^{m} \left( \hat{\alpha}_i - \alpha_i \right) y_i - \sum_{j=1}^{m} \left( \hat{\alpha}_j + \alpha_j \right) \varepsilon - \frac{1}{2} \sum_{i=1}^{m} \sum_{j=1}^{m} \left( \hat{\alpha}_i - \alpha_i \right) \left( \hat{\alpha}_j - \alpha_j \right) K \left( x_i, x_j \right), \tag{20}$$

where $\alpha_i, \hat{\alpha}_i, \alpha_j, \hat{\alpha}_j$ is the Lagrange multiplier and $K \left( x_i, x_j \right)$ is the kernel function. *Liu, Peng & Zheng (2021)* selects the RBF kernel function and obtains better prediction results. So the RBF kernel function used in this work is expressed as follows:

$$K \left( x_i, x_j \right) = \exp \left( -\frac{\| x_i - x_j \|^2}{2g^2} \right), \tag{21}$$

where $g$ is the kernel function parameter. The following final regression model of SVM can be obtained when Formula (20) is solved:

$$f \left( x, \hat{\alpha}_i, \alpha_i \right) = \sum_{i=1}^{m} \left( \hat{\alpha}_i - \alpha_i \right) K \left( x_i, x_j \right) + \hat{b}. \tag{22}$$

Among the variables, $\omega$ and $b$ are obtained during the operation. Values of the penalty coefficient $C$ and the kernel function parameter $g$ exert the maximum impact on the training accuracy of the support vector machine. The artificial bee colony algorithm will be used to optimize it, and the loss function uses the default value of 0.1.

## EXAMPLE ANALYSIS

### Dataset description
The study uses a one-year power load dataset in Nanjing, China. The data set includes power load, wind direction, wind power, maximum temperature, minimum temperature, weather conditions, whether the day is a holiday, and whether the day is a workday. The sampling interval of the influencing factor data is 1 day. The sampling interval of load data is 15 min.

### Data preprocessing and performance evaluation indicators
#### Data preprocessing
(1) Raw data missing value padding

In the process of raw data collection, individual data is inevitably missing. To reduce the impact of missing data on the accuracy of short-term load forecasting, it is necessary to fill in the missing raw information before predicting. There are many filling methods, such as 0 filling, mean filling, median filling, etc. In this study, the average value is used to serve. That is, the average value of the load before and after the missing value is used as the filling data to make it as close to the actual value as possible.

(2) Data normalization

Historical data are normalized as follows to avoid errors due to different unit dimensions between influencing factors and loads:

$$X' = \frac{X - X_{\min}}{X_{\max} - X_{\min}}, \tag{23}$$

where $X'$ is the normalized value of historical data, $X$ is the historical data value, $X_{\max}$ is the maximum value of the load and each influencing factor in the historical data, and $X_{\min}$ is the minimum value of the load and each influencing factor in the historical data.

### Performance evaluation index

The study used mean absolute error (MAE), mean absolute percentage error (MAPE), and root mean square error (RMSE) as the evaluation indicators of prediction methods. MAE, MAPE, and RMSE are calculated as follows:

$$MAE = \frac{1}{N} \sum_{i=1}^{N} \left| y'_i - y_i \right|, \tag{24}$$

$$MAPE = \frac{1}{N} \sum_{i=1}^{N} \left| \frac{y'_i - y_i}{y_i} \right| \times 100, \tag{25}$$

$$RMSE = \sqrt{\frac{1}{N} \sum_{i=1}^{N} (y'_i - y_i)^2}, \tag{26}$$

where $y'_i$ is the predicted load value and $y_i$ is the actual load value.

## Identification of important influencing factors

Weather conditions, wind direction, and holidays in influencing factors are usually expressed in Chinese characters and unrelated to the load. Therefore, quantization processing is required. According to the quantitative standard of *Zou et al. (2022)*, Monday to Friday are set as workdays and quantified as 1, and Saturday and Sunday are set as nonworkdays and quantified as 0. Similarly, the quantitative representation of wind direction, weather conditions, and whether the day is a holiday is listed in Table 3.

Let the influencing factor data be $X$, and set the matrix $X = [X_1, X_2, X_3, X_4, X_5, X_6]$, where $X_1$ is the wind direction, $X_2$ isthe minimum temperature, $X_3$ is the maximum temperature, $X_4$ is a workday, $X_5$ is a holiday, and $X_6$ is the weather condition. Let the load data be $Y$. Python is used to analyze the Pearson correlation coefficient between each influencing factor and the load and obtain the Pearson correlation coefficient value between each influencing factor and the load in Table 4.

As shown in Table 4, the absolute value of the Pearson correlation coefficient is the lowest temperature among the six influencing factors at 0.786. The minimum temperature exerts the maximum influence on the load, and the absolute value of the Pearson correlation coefficient is the weather condition at 0.026. Hence, weather conditions exert the minimum effect on the load. The influence of wind direction on the load is less than that of other

**Table 3  Weather condition, wind direction, and date type quantitative code.**

| Influencing factors | Coding | Influencing factors | Coding |
|---|---|---|---|
| Holiday | 1 if the day is a holiday and 0 if otherwise | West wind | 1.75 |
| Weekend | 0 | Northwest wind | 1.875 |
| Workday | 1 | Sunny | 2 |
| North wind | 1 | Partly cloudy | 1 |
| Northeasterly wind | 1.125 | Cloudy day | 0 |
| East wind | 1.25 | Light rain or snow | −1 |
| Southeast wind | 1.375 | Moderate rain or moderate snow | −2 |
| South wind | 1.5 | Heavy rain or snow | −3 |
| Southwesterly wind | 1.625 | Heavy rain or blizzard | −4 |

**Table 4  Pearson correlation coefficient of influencing factors.**

| Influencing factors | Coefficients | Influencing factors | Coefficients |
|---|---|---|---|
| Wind direction | 0.131 | Is the day a workday? | 0.537 |
| Minimum temperature | 0.786 | Is the day a holiday? | −0.584 |
| Maximum temperature | 0.756 | Weather conditions | 0.026 |

factors. Therefore, wind direction and weather conditions are considered weak influencing factors that will be ignored in this study. The four external factors of minimum temperature, maximum temperature, whether the day is a workday, and whether the day is a holiday are important influencing factors for selecting similar days.

## Similar day selection

August 31, 2003 is used as the date to be forecasted in this section. Data of influencing factors between the date to be forecasted and each historic day are extracted, and Python is utilized for gray correlation analysis and calculation of the gray correlation degree between each historical day and the day to be forecasted (Fig. 6). Because the Pearson correlation coefficient between the lowest temperature and the highest temperature is the largest, and it occupies the largest weight in the selection of similar days, so from the time scale, the closer the historical day to the day to be predicted, the greater the gray correlation value. *Huang et al. (2021)* sets the threshold to 0.7 and achieves better prediction results. Therefore, the threshold is set at 0.7, and the historical days greater than 0.7 are taken to form a rough set of similar days, including a total of 134 historical days.

Influencing factor data of each historic day in the rough set of the day to be predicted and similar days are extracted as the input feature dimension according to the rough set of similar days. Python is applied to perform $K$-means clustering. $K$-means clustering divides the historical days in the rough set of similar days into different categories according to the number of cluster centres. It obtains the coordinate values of the centre points of each type. Calculate the Euclidean distance between each influencing factor data and the coordinate value of each centre point on the day to be predicted, and the historical day included in the category with the smallest Euclidean distance from the day to be expected

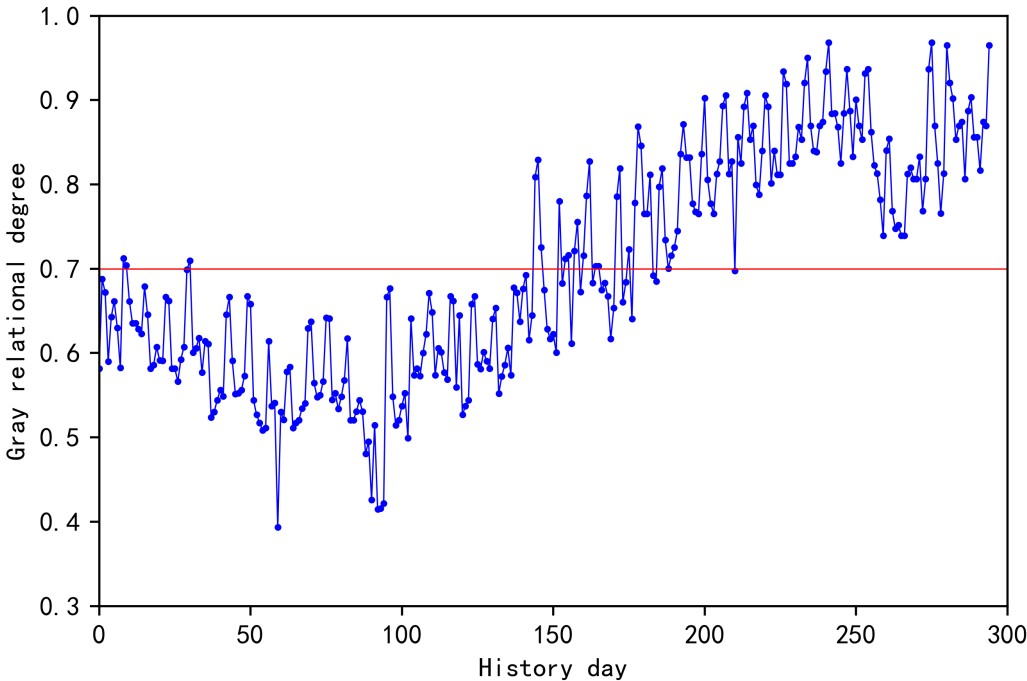

**Figure 6** Gray correlation between days to be predicted and historical days.

**Table 5** Euclidean distance between the day to be predicted and the cluster center.

| Category | Category 1 | Category 2 |
|---|---|---|
| Euclidean distance | 2.85 | 10.78 |

is the final set of similar days. The number of cluster centers will affect the clustering effect, and the silhouette coefficient (SIL) is an essential indicator for judging the number of cluster centers. The relationship between the number of cluster centers and the silhouette coefficient is shown in Fig. 7. The clustering effect enhances as the silhouette coefficient approaches 1. Therefore, the number of cluster centers is determined as 2.

The clustering effect of each historic day in the rough set of similar days is presented in Fig. 8. A total of 134 data points exist, and some data points are covered due to overlapping. Table 5 shows the Euclidean distance between the influencing factors on the day to be predicted and each cluster center. It can be seen from Table 5 that the Euclidean distance between the day to be predicted and the cluster center of the first type is small, and the cluster where the cluster center is located contains a total of 88 historical days. Therefore, the set of these 88 historical days is taken as the final set of similar days.

## Validity analysis of the selection of similar days

A total of 296 historical days in the original data, 134 historical days in the rough set of similar days, and 88 historical days in the collection of similar days are used as input data to verify the validity of the selection of similar days. The top 80% of the historical data

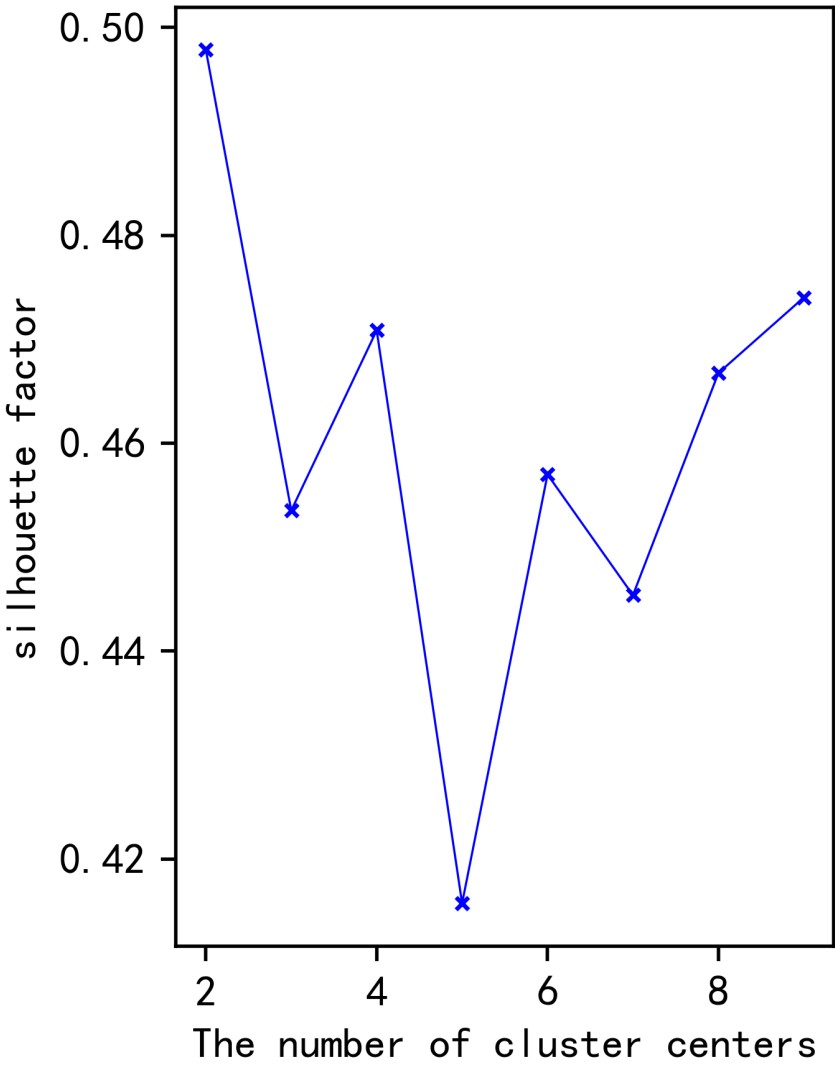

**Figure 7** Relationship between the number of cluster centers and the silhouette coefficient.

were selected. The daily data is used as the training set, and 20% of the historical daily data is used as the test set, which is input into the SVM prediction model after the parameters are optimized by the grid search method (*Jiang et al., 2018*) for training and testing. The penalty coefficient $C$ of SVM is 0.8, and the kernel function parameter $g$ is 100.

Three methods are programmed and case analysis is conducted in this study. (1) Method 1: Utilize Pearson correlation coefficient analysis to screen the influencing factors, use the original data as input data, apply the grid search method to optimize SVM parameters, establish the optimized SVM model, and record the method as SVM. (2) Method 2: Utilize Pearson correlation coefficient analysis to filter influencing factors, adopt the gray correlation analysis method to construct the similar daily rough set, use the similar daily rough set as the input data, apply the grid search method to optimize SVM parameters, establish the optimized SVM model, and denote the method as GSVM. (3) Method 3:

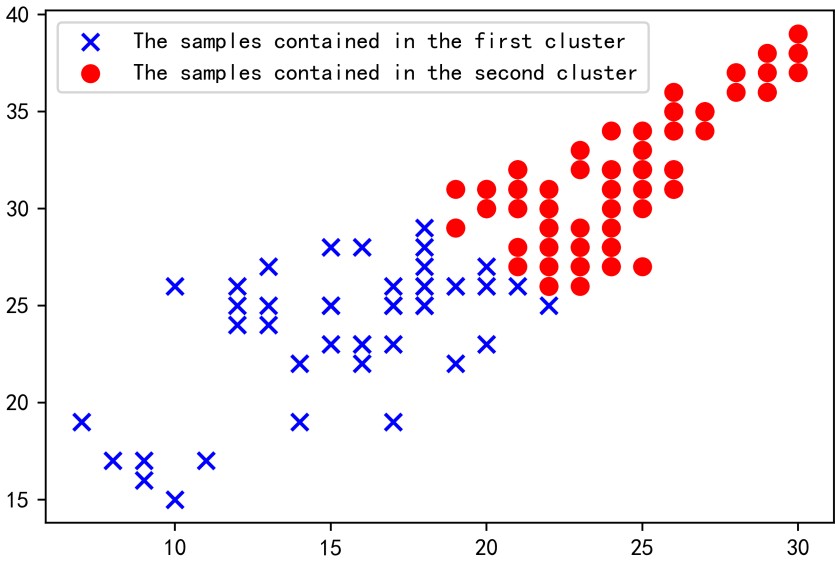

**Figure 8** Clustering effect diagram of each historical day in the rough set of similar days.

**Table 6  MAE, MAPE, and RMSE values for the three methods.**

| Method of prediction | MAE(MW) | MAPE(%) | RMSE(MW) |
| --- | --- | --- | --- |
| GSVM | 84.46 | 4.38 | 106.69 |
| GKSVM | 73.97 | 4.01 | 83.52 |
| SVM | 119.25 | 6.12 | 152.47 |

Use Pearson correlation coefficient analysis to filter influencing factors, utilize the gray correlation analysis method to construct a rough set of similar days, apply $K$-means clustering to construct a set of similar days further, use the set of similar days as input data, use the grid search method, optimize SVM parameters, establish the optimized SVM model, and record the method as GKSVM.

Errors of the three methods are illustrated in Fig. 9. MAE, MAPE, and RMSE values for the three methods are listed in Table 6. SVM shows the maximum improvement in prediction accuracy when similar days are selected twice, and the MAE, MAPE, and RMSE values are the minimum. The prediction accuracy of SVM without a similar date and time is the minimum, and the MAE, MAPE and RMSE values are the maximum. Therefore, the prediction accuracy will reduce when the original data contains a large amount of daily historical data unrelated to the date to be predicted. This finding proves the effectiveness of the proposed method. The load prediction values of the three methods are shown in Fig. 10. The results showed that the proposed method achieves the maximum degree of coincidence with the actual value.

## Artificial bee colony algorithm for optimizing SVM parameters

The artificial bee colony algorithm is used to optimize parameters $C$ and $g$ because the grid search method fails to find optimal parameters $C$ and $g$ of the support vector machine to

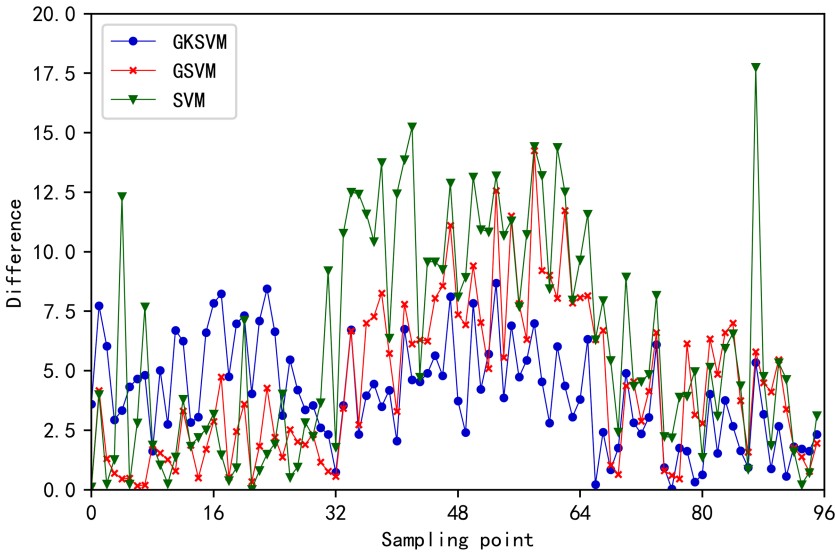

**Figure 9** Predicted load error value at each moment for the three methods.

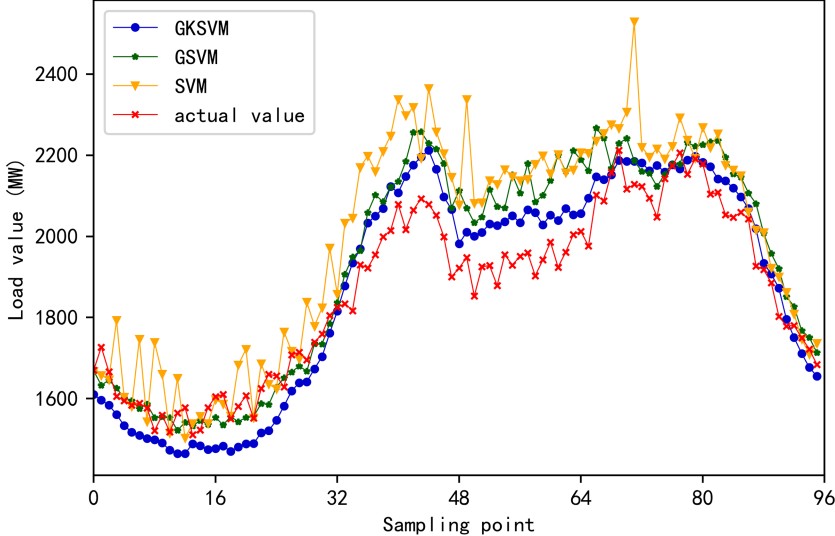

**Figure 10** Comparison chart of predicted and actual values of 96 sampling points for the three methods.

improve the prediction accuracy of the short-term power load. The set of similar days is used as input data, the population size of the ABC algorithm is 20, the number of nectar sources is $SN = 10$, the maximum number of iterations is $MCN = 100$, the maximum number of cycles of nectar sources is limited to 100, the SVM penalty coefficient is $C \in [0.01, 50]$, and the kernel function parameter is $g \in [0.01, 50]$. The mean square error (MSE) between the predicted load value of SVM and the actual load value is utilized as the objective function $J$.

**Table 7  Optimal parameters *C* and *g* of the support vector machine at each moment.**

| Point in time | *C* | *g* | Point in time | *C* | *g* | Point in time | *C* | *g* | Point in time | *C* | *g* |
|---|---|---|---|---|---|---|---|---|---|---|---|
| 1 | 0.01 | 0.01 | 25 | 2.05 | 0.01 | 49 | 0.43 | 0.01 | 73 | 1.71 | 0.01 |
| 2 | 2.74 | 0.01 | 26 | 0.16 | 0.01 | 50 | 0.01 | 0.42 | 74 | 0.74 | 0.01 |
| 3 | 0.01 | 0.24 | 27 | 25 | 0.01 | 51 | 0.27 | 0.01 | 75 | 0.42 | 0.01 |
| 4 | 0.01 | 0.01 | 28 | 6.69 | 0.01 | 52 | 0.24 | 0.01 | 76 | 1.04 | 0.01 |
| 5 | 0.01 | 0.25 | 29 | 2.86 | 0.01 | 53 | 0.68 | 0.01 | 77 | 0.01 | 0.25 |
| 6 | 0.13 | 0.01 | 30 | 0.01 | 37.04 | 54 | 1.77 | 0.01 | 78 | 0.01 | 16.9 |
| 7 | 0.01 | 50 | 31 | 3.7 | 0.01 | 55 | 0.77 | 0.01 | 79 | 0.01 | 0.01 |
| 8 | 0.23 | 0.01 | 32 | 0.01 | 46.23 | 56 | 0.01 | 0.16 | 80 | 0.01 | 0.01 |
| 9 | 0.01 | 0.01 | 33 | 0.03 | 0.01 | 57 | 0.01 | 0.33 | 81 | 46.37 | 25.86 |
| 10 | 0.01 | 0.56 | 34 | 0.9 | 0.01 | 58 | 0.1 | 0.01 | 82 | 0.12 | 0.01 |
| 11 | 0.01 | 0.01 | 35 | 4.18 | 0.01 | 59 | 0.37 | 0.01 | 83 | 0.01 | 0.01 |
| 12 | 0.01 | 12.29 | 36 | 1.81 | 0.01 | 60 | 0.01 | 0.87 | 84 | 0.01 | 0.01 |
| 13 | 0.01 | 15.43 | 37 | 1.49 | 0.01 | 61 | 0.01 | 0.64 | 85 | 0.01 | 0.01 |
| 14 | 0.01 | 0.1 | 38 | 9.55 | 0.01 | 62 | 0.48 | 0.01 | 86 | 0.01 | 0.01 |
| 15 | 0.04 | 0.01 | 39 | 1.19 | 0.01 | 63 | 0.01 | 0.34 | 87 | 2.05 | 0.01 |
| 16 | 0.01 | 9.66 | 40 | 0.77 | 0.01 | 64 | 0.05 | 0.01 | 88 | 0.01 | 0.01 |
| 17 | 3.02 | 0.01 | 41 | 1.25 | 0.01 | 65 | 0.01 | 0.45 | 89 | 0.01 | 0.01 |
| 18 | 2.86 | 0.01 | 42 | 0.31 | 0.01 | 66 | 0.01 | 0.34 | 90 | 0.01 | 0.01 |
| 19 | 0.01 | 0.57 | 43 | 0.01 | 0.33 | 67 | 0.01 | 0.01 | 91 | 0.01 | 0.03 |
| 20 | 0.01 | 22.33 | 44 | 0.27 | 0.01 | 68 | 0.01 | 0.07 | 92 | 0.01 | 0.01 |
| 21 | 4.2 | 0.01 | 45 | 2.37 | 0.01 | 69 | 0.14 | 0.01 | 93 | 0.01 | 0.01 |
| 22 | 0.01 | 50 | 46 | 0.28 | 0.01 | 70 | 0.01 | 0.01 | 94 | 0.01 | 0.01 |
| 23 | 8.21 | 0.01 | 47 | 0.61 | 0.01 | 71 | 0.21 | 0.01 | 95 | 0.01 | 0.01 |
| 24 | 14.88 | 0.01 | 48 | 0.01 | 32.57 | 72 | 0.56 | 0.01 | 96 | 0.01 | 26.21 |

Ninety-six support vector machine prediction models are designed given that the relationship between load values corresponding to 96 sampling points in a day and influencing factors is different. Meanwhile, the ABC algorithm is used to optimize the penalty coefficient *C* and kernel function parameters *g* of 96 vector machines. Optimal parameters *C* and *g* of the SVM at each sampling point are listed in Table 7. Iterative curves of the artificial bee colony, PSO (particle swarm optimization) algorithms and GWO (grey wolf optimization) algorithm are shown in Fig. 11. Convergence times of the artificial bee colony algorithm are about 22, and the objective function value finally converges to 36. Convergence times of the particle swarm algorithm are approximately 60, and the objective function value finally converges to 50. The number of convergence times of the grey wolf optimization algorithm is about 35 times, and the objective function value finally converges to 40 times. Convergence times and the final convergence value of the artificial bee colony algorithm are better than those of the particle swarm algorithm and the grey wolf optimization algorithm.

Optimal solutions of the ABC, PSO and GWO algorithms were collected after running them 20 times to compare the performance of the three algorithms further. Relative

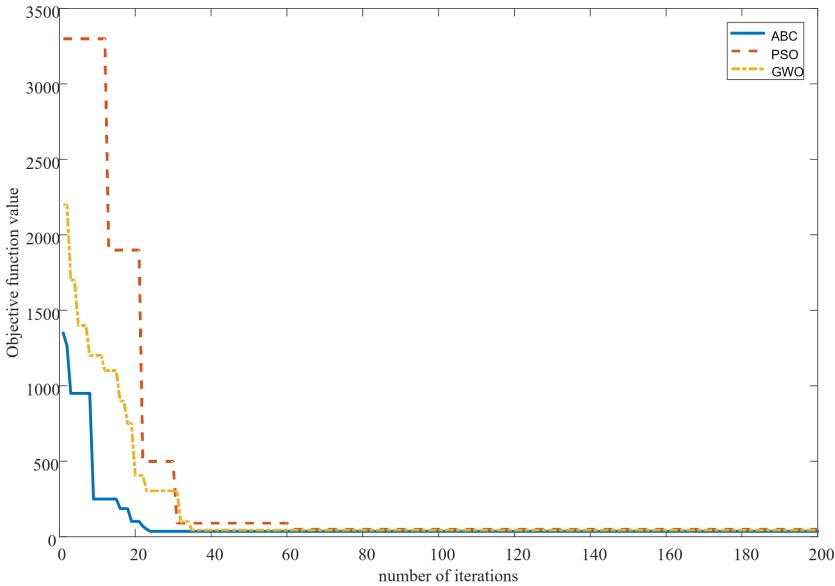

**Figure 11  ABC, PSO and GWO iteration curves.**

**Table 8  RPI values of ABC, PSO and GWO.**

| Number of runs | ABC | PSO | GWO | Number of runs | ABC | PSO | GWO |
|---|---|---|---|---|---|---|---|
| 1 | 1.772 | 25.221 | 12.292 | 11 | 0.332 | 27.575 | 12.569 |
| 2 | 0.249 | 24.197 | 9.275 | 12 | 2.187 | 20.903 | 11.157 |
| 3 | 3.848 | 23.976 | 10.382 | 13 | 1.467 | 22.841 | 9.192 |
| 4 | 1.301 | 28.820 | 11.822 | 14 | 1.800 | 23.007 | 8.195 |
| 5 | 0.554 | 26.495 | 10.410 | 15 | 0.305 | 28.239 | 12.209 |
| 6 | 4.845 | 22.149 | 13.372 | 16 | 2.298 | 27.658 | 12.514 |
| 7 | 3.3225 | 22.34 | 12.348 | 17 | 0.581 | 23.062 | 10.354 |
| 8 | 0.471 | 24.391 | 12.431 | 18 | 0.720 | 23.643 | 10.105 |
| 9 | 1.080 | 24.668 | 8.306 | 19 | 0 | 21.733 | 11.988 |
| 10 | 2.132 | 25.831 | 11.101 | 20 | 1.772 | 24.972 | 12.652 |

percentage growth rate ($RPI$) is used to evaluate the performance of the three algorithms as follows:

$$RPI(f) = \frac{(f - f^*)}{f^*} \times 100, \tag{27}$$

where $f$ represents the minimum mean square error between predicted and actual values of the support vector machine after a single operation of each algorithm and $f^*$ represents the minimum mean square error among all the minimum mean square errors. The $RPI$ values of the three algorithms ABC, PSO and GWO are shown in Table 8.

Optimal parameters obtained by the artificial bee colony algorithm demonstrate a more significant improvement in the prediction accuracy of the support vector machine

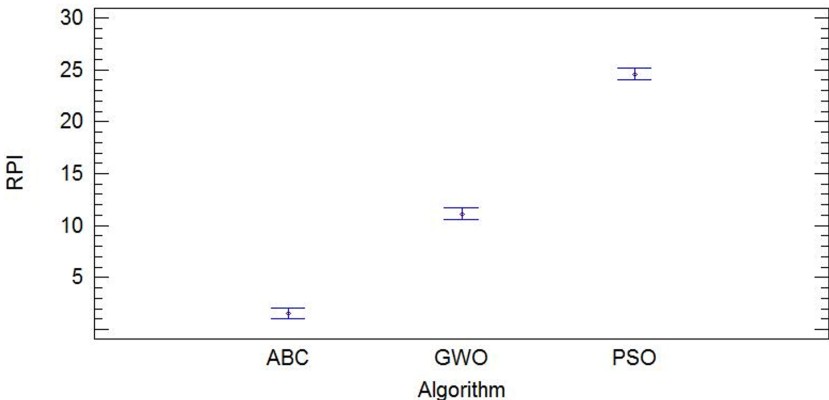

**Figure 12** Multivariate ANOVA of ABC, PSO and GWO.

compared with those of the particle swarm optimization algorithm and the grey wolf optimization algorithm. Multivariate analysis of variance (ANOVA) was performed, in which ABC, PSO and GWO were defined as factors, to show the performance gap between ABC, PSO and GWO intuitively. The results with a confidence level of 95% are shown in Fig. 12.

## Analysis of prediction results

The five methods are programmed and numerical example analysis is conducted in this study. (1) Method 1: Use the set of similar days as input data, optimize SVM parameters via the grid search method, build the optimized SVM model, and record the method as GKSVM. (2) Method 2: Use the set of similar days as input data, optimize SVM parameters through the particle swarm algorithm, build the optimized SVM model, and record the method as PSO-GKSVM. (3) Method 3: Take the set of similar days as the input data, use the grey wolf optimization algorithm to optimize the SVM parameters, build the optimized SVM model, and record the method as GWO-GKSVM. (4) Method 4: Use the set of similar days as input data to build an LSTM model, and this method is recorded as LSTM. (5) The proposed method in this study: Use the set of similar days as input data, utilize the ABC algorithm to optimize SVM parameters, establish the SVM model, and record the method as ABC-GKSVM.

The MAE, MAPE, and RMSE values of the five methods are listed in Table 9. The error between predicted and actual load values at 96 time points on August 31, 2003 is shown in Fig. 13. Figure 14 illustrates the comparison of predicted and actual values of the five methods at 96 time points. The scatter plot in Fig. 15 depicts the degree of agreement between predicted values of the five methods and actual values. The diagonal line of $y = x$ in the figure denoted that the predicted value is equal to the actual value. The point in the

**Table 9   MAE, MAPE, and RMSE values for the five methods.**

| Method of prediction | MAE(MW) | MAPE(%) | RMSE(MW) |
|---|---|---|---|
| GKSVM | 73.23 | 4.01 | 83.20 |
| PSO-GKSVM | 42.27 | 2.15 | 62.84 |
| GWO-GKSVM | 38.79 | 1.96 | 54.32 |
| LSTM | 83.42 | 4.48 | 87.58 |
| ABC-GKSVM | 34.98 | 1.79 | 46.38 |

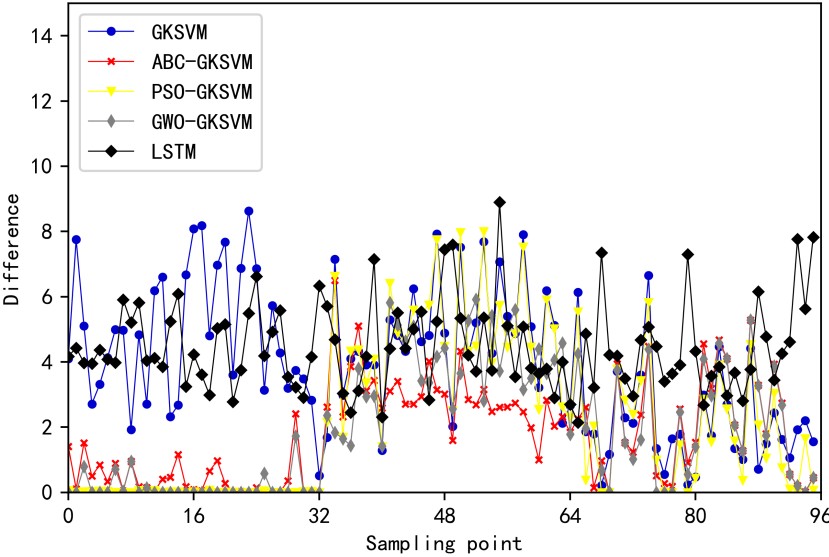

**Figure 13   Comparison chart of error values at each moment of the five methods.**

figure is close to the diagonal line when the difference between predicted and actual values is negligible.

Figures 13, 14, 15 and Table 9 showed that the points of ABC-GKSVM are generally the closest to the diagonal among the four methods when two similar days are selected; abnormal points are absent; and the MAE, MAPE, and RMSE values are all less than those of the support vector machine model optimized via the grid search method and particle swarm optimization algorithm. The consistency between the curve trend and actual values verified the effectiveness of the ABC algorithm in optimizing SVM parameters. Meanwhile, compared with the LSTM prediction model, the proposed method reduces the MAE, MAPE, and RMSE indicators by 48.44 MW, 2.69%, and 41.2 MW, respectively, thereby confirming the effectiveness of the proposed method further.

## CONCLUSIONS

Gray relational analysis and *K*-means clustering are used in this study to screen 298 historical data. A set of similar days is formed as input data after eliminating historical data that are irrelevant or slightly relevant to the day to be predicted. The ABC algorithm

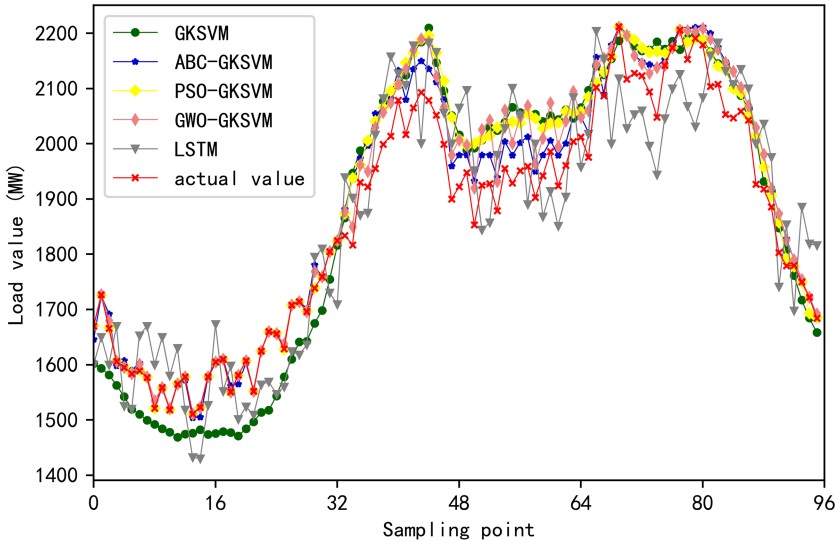

**Figure 14** Comparison of predicted and actual values for each method.

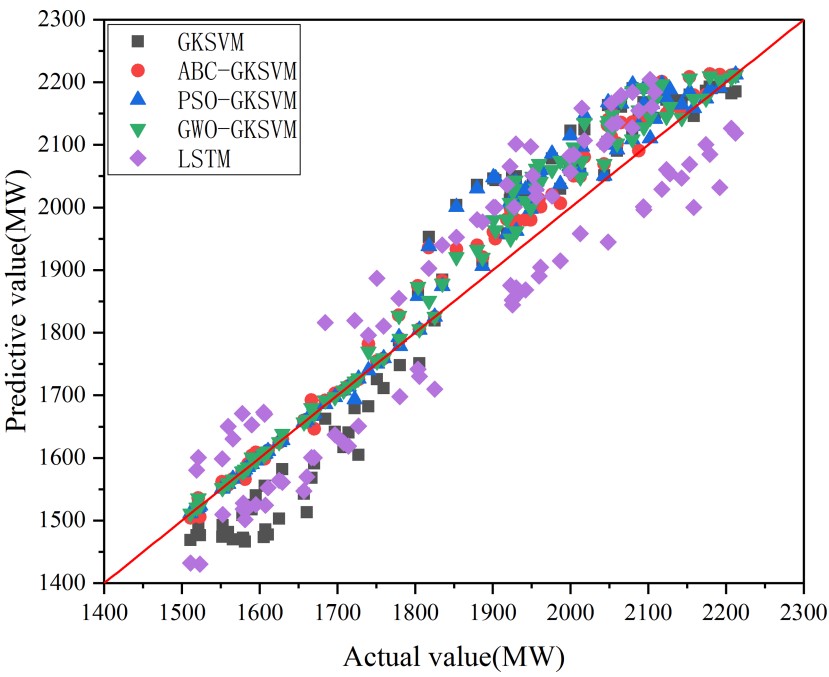

**Figure 15** Scatter plot of predicted values versus actual values for the five methods.

optimizes parameters $C$ and $g$ of the SVM, establishes the ABC-SVM short-term power load forecasting model, and finally obtains forecast values 96 times for the day to be forecasted. Furthermore, the effectiveness of the proposed method is proven using examples.

(1) The Pearson correlation coefficient method is used to calculate the Pearson coefficient value between each external factor and the load. External factors that significantly influence the load are screened according to the coefficient value.

(2) The gray correlation degree between the day to be predicted and each historical day is calculated through gray correlation analysis, and rough sets of similar days are filtered by setting the threshold. $K$-means clustering is utilized to classify the rough sets of similar days. The cluster center with the minimum Euclidean distance to each influencing factor in the days to be predicted is set as the set of similar days to reduce the input of irrelevant data.

(3) The artificial bee colony algorithm is applied to find the optimal penalty coefficient $C$ and kernel function parameter $g$ of the support vector machine suitable for the data of this study and address the problem of decreasing prediction accuracy of the support vector machine due to improper parameter selection.

The selection of external factors with a significant impact on the load and an hourly granularity can be the focus of future investigations to improve the prediction accuracy further given that the influence of temperature and weather conditions on the load contains hysteresis and cumulative effects.

### Funding

This work was supported by the National Natural Science Foundation of China (No. 61773269), the Natural Science Foundation of Liaoning Province of China (No. 2019-KF-03-08), the Department of Education of Liaoning Province of China (No. LJKZ1110), and the Program for Shenyang High Level Innovative Talents (No. RC190042). There was no additional external funding received for this study. The funders had no role in study design, data collection and analysis, decision to publish, or preparation of the manuscript.

### Grant Disclosures

The following grant information was disclosed by the authors:
National Natural Science Foundation of China: 61773269.
Natural Science Foundation of Liaoning Province of China: 2019-KF-03-08.
Department of Education of Liaoning Province of China: LJKZ1110.
Program for Shenyang High Level Innovative Talents: RC190042.

### Competing Interests

Changfeng Luan is employed by Yingkou Power Supply Company, State Grid Liaoning Electric Power Co., Ltd.

### Author Contributions

- Xinfu Pang conceived and designed the experiments, performed the experiments, analyzed the data, performed the computation work, authored or reviewed drafts of the article, and approved the final draft.

- Wei Sun conceived and designed the experiments, performed the experiments, analyzed the data, performed the computation work, prepared figures and/or tables, authored or reviewed drafts of the article, and approved the final draft.
- Haibo Li conceived and designed the experiments, performed the experiments, analyzed the data, authored or reviewed drafts of the article, and approved the final draft.
- Yibao Wang performed the computation work, prepared figures and/or tables, and approved the final draft.
- Changfeng Luan analyzed the data, authored or reviewed drafts of the article, and approved the final draft.

## Data Availability

The raw data and codes of GRA, K-means clustering, SVM, PSO, ABC, GWO, and LSTM are available in the Supplemental Files.

## Supplemental Information

Supplemental information for this article can be found online at http://dx.doi.org/10.7717/peerj-cs.1108#supplemental-information.

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
