# Peer review of "Short-term power load forecasting based on gray relational analysis and support vector machine optimized by artificial bee colony algorithm"

_PeerJ Computer Science, doi:10.7717/peerj-cs.1108_

## Round 0.1 · original submission · Major Revisions

The paper is well written with publishable contributions. However, some minor issues should be taken into account before accepting it. Basically, the study investigates two methods GRA and K-means. The authors have not clearly highlighted the advantages and disadvantages of these two methods and why the study is around two methods. Generally speaking, we only focus on the most advantage method rather than comparing various methods, unless the method has its own benefit. The presentation should also be improved, such as the symbol and terminology explanations. For example, what is PSO short for? The authors should further check all the typos and grammar problem in the revision.

More importantly, we notice a paper related to the contents of this submission: Aranha, C., Camacho Villalón, C.L., Campelo, F. et al. Metaphor-based metaheuristics, a call for action: the elephant in the room. Swarm Intell 16, 1–6 (2022). https://doi.org/10.1007/s11721-021-00202-9. In which, the following 4 standards have been mentioned. Please address all the following points in the revision.

> (i) present their method using the normal, standard optimization terminology;
> (ii) show that the new method brings useful and novel concepts to the field;
> (iii) motivate the use of the metaphor on a sound, scientific basis;
> (iv) present a fair comparison with other state-of-the-art methods using state-of-the-art practices for benchmarking algorithms.

In summary, a major correction is needed to improve the quality of the manuscript.

Reviewer 1 ·

Basic reporting

(1) Figure 13 is not clear enough, it is recommended to improve the clarity.
(2) Figure 14 has the same problem with before.

Experimental design

(1) The study uses GRA and K-means clustering to select two similar days. What is the advantage of two similar days selection over one similar day selection? No description is seen.
(2) What preprocessing was performed on the original data during the example simulation? No description is seen.
(3) What is the quantitative basis for the wind direction, whether a working day or a holiday and the weather when the article conducts the Pearson correlation coefficient analysis?

Validity of the findings

(1) When using LSTM neural network for prediction, is the input data selected after two similar day selections? This has an impact on the validity analysis of the experimental results.

Additional comments

(1) In the process of SVM training, what is the basis for selecting the SVM kernel function? No description is seen.
(2) How are insensitive loss factors valued? How does the loss factor affect the accuracy of power load forecasting?

Annotated reviews are not available for download in order to protect the identity of reviewers who chose to remain anonymous.

Reviewer 2 ·

Basic reporting

This paper proposed a short-term load forecasting method, which is based on the gray relational analysis and artificial bee colony-support vector machine algorithm. The methods are applied to the case study of load forecasting and verified the effectiveness. The paper is well-written, however, the following points need to be clarified:

Experimental design

1. Page 10, K-means clustering method is used to find the sets of similar days. But this method applied to the short-term load forecast is not clearly explained.
2. Similarly, the gray correlation analysis is used to get the correlation between the predicted days and historical ones. So why are the gray correlation analysis and K-means methods both applied, since they have similar functions?

Validity of the findings

3. Page 17, what is PSO short for? Particle swan algorithm? Figure 11 and Table 8 compare the performance of the two methods. But the PSO is a relatively traditional method. Other advanced methods for forecasting should be compared with the proposed method and the computational efficiency should be compared.

Reviewer 3 ·

Basic reporting

the paper contain the proper references to existing literature, and is sufficiently concise and organized.

Experimental design

the experimental methods are described clearly and with sufficient details.

Validity of the findings

the author`s conclusions are justified by the data. The
statistical treatment of the data is adequate, and the data is understandable.

Additional comments

The article meets the PeerJ criteria and should be accepted as is.

---

## Round 0.2 · accepted · Accept

All the concerns have been addressed well in the revised version and all the reviewers satisfy the revision. The authors have clearly explained the requested 4 standards in the editor's letter. Since no further comments are received, I recommend accepting this manuscript as it is based on solid contributions.

[# Reviewer 1 ·

Basic reporting

This section has been fully revised in accordance with my previous comments

Experimental design

This part has been fully revised in accordance with my previous comments

Validity of the findings

This section has been fully revised aacording to my suggestion.

Additional comments

This paper has been fully revised in accordance with my previous opinions, and I suggest acceptance .

Reviewer 2 ·

Basic reporting

Questions are answered, no more comments

Experimental design

Questions are answered, no more comments

Validity of the findings

Questions are answered, no more comments

Additional comments

Questions are answered, no more comments